# Factors associated with malaria in indigenous populations: A retrospective study from 2007 to 2016

Bruna Martins Meireles[1], Vanderson de Souza Sampaio[2,3,4], Wuelton Marcelo Monteiro[2,3], Maria Jacirema Ferreira Gonçalves[1,5¤]*

1 Escola de Enfermagem de Manaus, Universidade Federal do Amazonas, Manaus, Amazonas, Brazil, 2 Fundação de Medicina Tropical Dr. Heitor Vieira Dourado, Manaus, Amazonas, Brazil, 3 Universidade do Estado do Amazonas, Manaus, Amazonas, Brazil, 4 Fundação de Vigilância em Saúde do Estado do Amazonas, Manaus, Amazonas, Brazil, 5 Instituto Leônidas & Maria Deane/Fiocruz, Manaus, Brazil

¤ Current address: Universidade Federal do Amazonas, Manaus, Amazonas, Brazil
* jaciremagoncalves@gmail.com

## Abstract

### Background

In Brazil malaria is most frequent in the Amazon region, mainly in the Amazonas state, where it is found the most proportion of indigenous people of the whole country. It is remarkable publications about malaria in the Amazon, although information on malaria in indigenous populations is still poorly explored.

### Objective

Identify factors associated with malaria in indigenous populations.

### Methods

Cross-sectional study of positive cases of malaria in the state of Amazonas, Brazil, from 2007 to 2016. Secondary data were obtained from the Epidemiological Surveillance Information System for Malaria and from the Mortality Information System, both from Brazil. To tackle with race missing data, cases with no race fulfilled were classified according to the probable location where infection occurred. This way, was imputed indigenous race for those which the probable infection location was indigenous village (aldeia). Variables tested with race were: sex, age, schooling, microscope surveillance slide type, parasitic infection species, parasitemia level, and timeliness of treatment. Multivariate logistic regression was used.

### Results

A total of 1,055,852 cases of malaria were notified in the state of Amazonas. Among the factors that associate malaria and indigenous peoples, the most significant were sex, children and high levels of parasitemia. The magnitude of *Plasmodium vivax* infection is higher than

**Data Availability Statement:** The data underlying the findings cannot be made publicly available in order to protect patient privacy and comply with Brazilian law. Data from notification malaria cases

can be obtained upon request from Foundation of Health Surveillance (Fundação de Vigilância em Saúde) of the Health State Secretariat of Amazonas, Manaus, AM, Brazil. Readers may contact Dr. Rosemary Costa Pinto (FVS-Amazonas) at neplaifvs@gmail.com or please contact the coordinator the Ethics Committee of the Amazonas Federal University Eliana Maria Pereira da Fonseca (cep.ufam@gmail.com).

**Funding:** Program to support the scientific publication article by the Amazonas State Research Support Foundation - Fundação de Amparo à Pesquisa do Estado do Amazonas (PAPAC / FAPEAM).

**Competing interests:** The authors have declared that no competing interests exist.

*Plasmodium falciparum*, although this parasite was more frequent in indigenous than other races. In regards to mortality, 109 deaths were registered, most of them related to *P. vivax*.

## Conclusion

The findings underscore the importance of look at indigenous people differently of other races. The associated factors highlight a profile of cases severity, because of highest parasitemia, many cases of *P. falciparum* although high frequency of *P. vivax*, and children. Furthermore, the mortality in indigenous, specially in older people is worrying

## Introduction

Malaria is a worldwide health problem, with the majority of cases occurring on the African continent (93%), followed by the Southeast Asian region (3.4%) and the Eastern Mediterranean region (2.1%). In 2018, it is estimated that there were 228 million cases of malaria worldwide, in comparison to 251 million cases in 2010, and 231 million cases in 2017. The incidence rate of malaria decreased globally between 2010 and 2018, from 71 to 57 cases per thousand at risk inhabitants [1]. Despite being avoidable and treatable, malaria continues to have a great impact on human health and its subsistence [1].

In the Americas, despite having a lower prevalence when compared to global levels, malaria is a severe problem. Three countries (Brazil, Colombia and Venezuela) account for 80% of all estimated cases in the Americas, which 51% of the cases occurred in Venezuela, followed by Brazil with 23% of the cases in 2018 [1]. In Brazil, the majority of cases (99%) are concentrated in the Amazon region, which is linked to environmental factors that favor the occurrence and transmission of malaria such as high temperatures, high precipitation and humidity, low altitude and its extensive network of rivers [2, 3]. Furthermore, environmental changes due to anthropic activity, especially Forest clearing, cause great impact on the malaria transmission. Such alterations favor vector breeding sites that, when combined with the population's lifestyle and its demographic and socioeconomic characteristics, impact on the occurrence of malaria. In this context, the indigenous population may be more vulnerable to malaria, due as much to the rural location of their homes as to their lifestyle [4].

The state of Amazonas reported more than 82,000 new cases of malaria in 2017 [5]. This state also has the highest proportional indigenous population in Brazil, as it has 183,514 indigenous inhabitants, which corresponds to 22.4% of the total of the country [6]. Although there are publications about malaria in the Amazon, information on malaria in indigenous populations is still poorly explored and reported in a generic manner [7–9]. Therefore, studies on the association between the occurrence of malaria and indigenous people, as well as the magnitude of the disease in this population, are necessary. The aim of this study was to the aim of this study was to identify factors associated with malaria in indigenous people, in a scenario of high malaria incidence in the state of Amazonas, Brazil, from 2007 to 2016.

## Materials and methods

### Study site and population

The study site is the state of Amazonas which, according to the 2010 Population Census, had a population of 3,483,985 inhabitants, in a total area of 1,559,161 km$^2$ and is comprised of 62 municipalities [10]. Amazonas is the Brazilian state with the most proportion of indigenous population (22.4%). Brazil counts 444,747 indigenous people, of which 88,299 are in the state

of Amazonas, and 70% live on indigenous lands. There are 2344 indigenous villages (aldeias) and 220 ethnic groups in the state of Amazonas.

The National Malaria Control Program in Brazil relies on the early diagnosis and treatment of malaria cases, whose infection is detected by microscopy. The treatment policy for uncomplicated malaria: infections by *P. vivax* or *P. malariae* is chloroquine in 3 days and primaquine in 7 days; infections by *P. falciparum*, the association of artemeter with lumefantrine is used [2].

As a demonstration of continued surveillance efforts to establish a process of expanding the supply of timeliness of diagnosis and treatment, regardless of the extensive network of malaria laboratories available throughout the Brazilian Amazon (3,492 laboratories and 48,849 healthcare professionals across the state of Amazonas network) [3], the use of Rapid Diagnostic Tests (RDT) is being encouraged in Brazil, especially in remote areas, where there is less access to microscopy facilities [2]. The distribution and use of RDTs increased from 1486 tests in 2011 to 14,655 in 2015, and occurred principally in areas without good microscopy facilities [2, 3], despite this, highest number of passive malaria case detection are carried out. There is no difference in the indigenous malaria health program to the general population in Brazil. After diagnosis of malaria, treatment is immediately prescribed and administered free of charge to all the patients.

This study uses the categories of race in regards to registered cases of malaria, as per the methodology of the Brazilian Institute of Geography and Statistics (IBGE) on self-reported race. However, the registration of this particular variable in the Epidemiological Surveillance Information System (SIVEP) was only mandatory from 2013 onwards, so an analysis of this nature needs to deal with missing data. From 2013 on, the following definitions were applied:

- Indigenous (Malaria in patients who identified themselves as indigenous): positive malaria cases in individuals who identified themselves as indigenous or those with indigenous villages as their likely place of infection, as reported on SIVEP.

- Other races (Non-indigenous): positive malaria cases involving white, black, yellow, brown-skinned patients or those who chose not to reply.

- No information regarding race (missing): cases in which the race variable was not filled in and cannot be automatically attributed because it was a case in which the place of probable infection was not the indigenous village.

## Study design

Cross-sectional study of positive cases of malaria in the state of Amazonas, from 2007 to 2016, using secondary data from national surveillance system.

## Ethical considerations

All ethical criteria related to Resolution 466/12 of the National Health Council were respected, especially regarding confidentiality and non-disclosure of information. This study was approved by the Ethics Review Board of the Federal University of Amazonas (Process Number 2,302,738—September 28, 2017), and requested a waiver of the Informed Consent Form, since it dealt with secondary data, systematically collected in the services and obtained without identifying the subjects.

## Procedures and sources of data

Reports of new positive cases of malaria were obtained from the Malaria Epidemiological Surveillance Information System (SIVEP-Malaria), excluding relapsing cases and treatment failure. Malaria death statistics were obtained from the Mortality Information System (MIS). Only the cases of malaria in which the patients' municipality of residence and infection report

came from the state of Amazonas, and which occurred between January 1, 2007 and December 31, 2016 were considered. The Ministry of Health malaria case definition is: malaria suspect person whose presence of parasite or some of its components have been identified in the blood by laboratory examination. Therefore, out of a total of 1,055,852 notifications, only 936,379 cases were considered in this analysis.

A total of 1,055,852 cases of malaria were reported between 2007 and 2016. Of these 119,473 (11.3%) were excluded due to lack of data, errors when registering the case or that slides registered just to review cured cases of those new cases already reported, which resulted in 936,379 cases analyzed. The completeness of the race variable was 40.2% throughout the period, which corresponded to 376,300 subjects with information on race. It is noteworthy that this variable was only mandatory from 2013 onwards, from whence onwards the completeness was 96.2% (2013–2016).

We considered indigenous people those who were this way filled in the race variable. Cases without race information were reclassified for the purpose of this study. So, we considered indigenous race if there was no race data and the probable place of infection was the indigenous village. Thus, 42,702 subjects, who previously had missing information on race, received an indigenous classification based on this criterion.

The option to use "probable place of infection = indigenous village" was a form of "multiple imputation". This interpretation was assumed based on the guidelines of SIVEP which defines a village (known as aldeia in Portuguese) as a "cluster of indigenous areas include villages and huts". Therefore, it is assumed that the records that had village (aldeia) as a probable place of infection refer to the occurrence of malaria in indigenous populations, since they are characteristic of indigenous peoples' dwellings, and that the non-indigenous population would be included as little as possible. This methodology for imputing indigenous race has a low risk of bias, since a subject whose likely infection site is the "village" must be a local resident, and therefore, indigenous. On the other hand, there are health professionals who access the villages, however, being a case of malaria, it is more likely that the race information will be filled in, otherwise, it can also be considered that the quantity of health professionals is small face to the local indigenous population, causing no impact on this analysis.

## Data analysis

The database was organized and analyzed using *Stata* software version 13.

For morbidity analysis, the cases were stratified into indigenous, non-indigenous and those without information about their race. The analysis of deaths included the registration of race in the MIS, which stratified it into indigenous and non-indigenous, since completeness of the information was 100% for this last data source.

In the descriptive phase, data were analyzed according to frequency and distribution, presenting categorical variables in percentages. For ordinal variables, the chi-square trend test was performed to verify the tendency of increases or reductions in proportions. P-value <0.01 was considered statistically significant.

For multivariate analysis, stepwise logistic regression was used to identify factors associated with malaria, and controlled for the other variables in each stratum: indigenous, and non-indigenous. Those cases without data on their race were analyzed descriptively.

In the modeling process the P-value significance criterion <0.20 was used to include the variables in the model, based on the bivariate analysis, and the variables with statistical significance, with P-value <0.05, were maintained in the final model. Association measures were calculated using crude and adjusted odds ratios (OR), together with their confidence intervals (95% CI). The fit of the model was evaluated by the Hosmer & Lemeshow test.

## Variables

The variables used were extracted from the databases (SIVEP and MIS) and the completeness of each variable was calculated.

The following variables were used:

a. Spatial: zone, municipality of probable infection and residence, place of probable infection and residence;

b. Demographics: sex (female and male); age group in years (<1, 1year of age <10, 10 years of age <20, 20 years of age <40, 40 years of age <60 and ≥60), race (indigenous, non-indigenous, no data on race) and schooling (years of study) (illiterate, 1–4 years of study, 5–9 years of study, ≥10 years of study);

c. Clinical-Laboratory: microscope surveillance slide type: active case surveillance (health professionals look for cases). After notification of one or more cases of malaria and determination of the place of transmission, professionals perform active detection of other symptomatic or non-symptomatic cases; passive case surveillance (when the patient, usually with symptoms, seeks the notifying health service to be tested)

   Parasitic infection species:

   1. *Plasmodium vivax;*

   2. *Plasmodium falciparum* which included *P. falciparum* and *P. falciparum* + gametocytes of *P. falciparum;*

   3. Mixed infection by *P. vivax* + *P. falciparum* which included *P. vivax* + gametocytes of *P. falciparum* and *P. falciparum* + *P. vivax;*

   4. Other species which included infections caused by *P.malariae* and *P. ovale.*

d. Parasitemia level in the positive cases (the number of parasites in thick smears) was graded using the plus system scale: <+/2 (less than one parasite per 100 thick film fields); +/2 (up to one parasite per 100 thick film fields); + (1 to 10 parasites per 100 thick film fields); ++ (11 to 100 parasites per 100 thick film fields); +++ (1 to 10 parasites per one thick film field); ++++ (>10 parasites per one thick film field);

e. Treatment effectiveness: time between first symptoms and diagnosis (days); Time between first symptoms and treatment (days); and timeliness of treatment (relationship between date of diagnosis and date of start of treatment).

## Results

Malaria cases were predominant in males in all categories, but in non indigenous people and missing race data this proportion was higher. In the indigenous category there was a predominance in the age group between 1 and 10 years of age, whose proportion decreases as age increases, while in non-indigenous and missing races groups the predominance occurred in the age range between 20 and <40 years of age. The low level of education was similar in both indigenous and non-indigenous groups, with higher proportions in individuals with only 1 to 4 years of schooling (Table 1).

The passive case detection was more prevalent in all race categories. Parasitemia <+/2 (less than one parasite per 100 thick film fields) showed a higher percentage in the non-indigenous group; while parasitemia of ++ (11 to 100 parasites per 100 thick film fields) was higher among the indigenous group and in the cases where race was unspecified. In all racial groups, the

**Table 1. Characterization of malaria cases by location of probable infection, in indigenous patients (village or self-reported indigenous), non-indigenous and missing data on race.** Amazonas state, Brazil, 2007–2016.

| Variables | Indigenous* | | Non-Indigenous** | | Missing data | |
|---|---|---|---|---|---|---|
| | N | % | n | % | n | % |
| **Sex** (completeness: >99%) | | | | | | |
| Male | 91680 | 54.30 | 151595 | 60.60 | 310436 | 60.00 |
| Female | 77150 | 45.70 | 98575 | 39.40 | 20694 | 40.00 |
| **Age group (years)**† (completeness: 100%) | | | | | | |
| < 1 | 4593 | 2.72 | 2990 | 1.20 | 7328 | 1.42 |
| 1 to < 10 | 60747 | 35.98 | 52393 | 20.94 | 122218 | 23.62 |
| 10 to < 20 | 40209 | 23.82 | 63472 | 25.37 | 117476 | 22.71 |
| 20 to < 40 | 41663 | 24.68 | 82076 | 32.81 | 168448 | 32.56 |
| 40 to < 60 | 15427 | 9.14 | 38354 | 15.33 | 80040 | 15.47 |
| 60 or over | 38937 | 3.67 | 10879 | 4.35 | 21867 | 4.23 |
| **Schooling (years)**† (completeness: 59.2%) | | | | | | |
| Illiterate | 38043 | 23.06 | 26260 | 23.06 | 56950 | 12.83 |
| 1 to 4 | 50579 | 30.65 | 95838 | 38.66 | 143898 | 32.41 |
| 5 to 9 | 23951 | 14.52 | 61357 | 24.75 | 152193 | 34.28 |
| 10 or more | 15055 | 9.12 | 37482 | 15.12 | 31438 | 7.08 |
| Not applicable | 37377 | 22.65 | 26948 | 10.87 | 59540 | 13.41 |
| **Type of surveillance slide** (completeness: 100%) | | | | | | |
| Passive case surveillance | 89460 | 52.99 | 172170 | 52.99 | 366908 | 70.92 |
| Active case surveillance | 79370 | 47.01 | 78002 | 31.18 | 150469 | 29.08 |
| **Parasitemia level by plus system scale** † (completeness: 99.5%) | | | | | | |
| < +/2 (less than one parasite per 100 thick film fields) | 41647 | 25.22 | 80912 | 32.51 | 138139 | 26.70 |
| +/2 (up to one parasite per 100 thick film fields) | 32112 | 19.45 | 41741 | 16.77 | 91971 | 17.78 |
| + (1 to 10 parasites per 100 thick film fields) | 40062 | 24.26 | 53708 | 21.58 | 118216 | 22.85 |
| ++ (11 to 100 parasites per 100 thick film fields) | 47607 | 28.83 | 68904 | 27.68 | 157327 | 30.41 |
| +++ (1 to 10 parasites per one thick film field) | 3600 | 2.18 | 3563 | 1.43 | 11399 | 2.20 |
| ++++ (>10 parasites per one thick film field) | 104 | 0.06 | 74 | 0.03 | 322 | 0.06 |
| **Test results** (completeness: 100%) | | | | | | |
| P. *vivax* | 145132 | 85.96 | 229533 | 91.75 | 447884 | 86.57 |
| P. *falciparum* | 22098 | 13.09 | 19811 | 7.92 | 66123 | 12.78 |
| Mixed infection | 1571 | 0.93 | 814 | 0.33 | 3366 | 0.65 |
| Other | 29 | 0.02 | 14 | 0.01 | 4 | 0.00 |
| **Time between first symptoms and diagnosis (days)** | | | | | | |
| 0 | 34713 | 20.56 | 32853 | 13.13 | 800335 | 15.47 |
| 1 | 35086 | 20.78 | 56764 | 22.69 | 112645 | 21.77 |
| 2 | 30083 | 17.82 | 51899 | 20.75 | 100055 | 19.34 |
| 3 | 21106 | 12.50 | 37779 | 15.10 | 79498 | 15.37 |
| 4 to 7 | 25756 | 15.26 | 44174 | 17.66 | 96754 | 18.70 |
| > 7 | 22070 | 13.07 | 26670 | 10.66 | 48349 | 9.35 |
| **Time between first symptoms and treatment (days)** | | | | | | |
| 0 | 29064 | 17.22 | 30023 | 12.00 | 63916 | 12.35 |
| 1 | 34322 | 20.33 | 55576 | 22.22 | 104069 | 20.12 |
| 2 | 30694 | 18.18 | 52015 | 20.79 | 92611 | 17.90 |
| 3 | 22097 | 13.09 | 38350 | 15.33 | 71389 | 13.80 |
| 4 to 7 | 28461 | 16.86 | 46003 | 18.39 | 88402 | 17.09 |
| > 7 | 24178 | 14.32 | 28173 | 11.26 | 96970 | 18.74 |

(*Continued*)

**Table 1.** (Continued)

| Variables | Indigenous* | | Non-Indigenous** | | Missing data | |
|---|---|---|---|---|---|---|
| | N | % | n | % | n | % |
| **Timeliness of treatment (hours)** | | | | | | |
| ≤ 48 | 74759 | 44.28 | 112564 | 44.99 | 256791 | 49.63 |
| > 48 | 94071 | 55.72 | 137608 | 55.01 | 260586 | 50.37 |

Source: Epidemiological Surveillance Information System for Malaria, data obtained January, 2018.

The total sample was 936,379 cases. The percentage is calculated in the column.

† Chi-square test for significant trend with P-value <0.01 for all variables, in Indigenous classification, Non-indigenous and those missing data on race.

* Indigenous is considered the combination of an individual with a declared indigenous race or having the village as a place of infection.

** Non-indigenous is the combination of all other races, excluding the indigenous race and missing data.

main type of plasmodium was *P. vivax*, the predominant form of malaria in the region (Table 1).

In all categories of race, there is a gradual decrease in the number of malaria cases as increases the number of days between the onset of symptoms and diagnosis. For the amount of time elapsed between the first symptoms and the beginning of treatment, the longest period observed in all categories was up to 2 days (Table 1).

Regarding timeliness of treatment of malaria, the cases registered in all races categories (indigenous, non-indigenous and missing data groups) presented timely treatment longer than 48 hours: indigenous (55.7%), non-indigenous (55.0%) and those without data on race (50.3%) (Table 1).

The result of logistic regression analysis in order to estimate the factors associated with indigenous malaria with the respective values of crude OR and adjusted OR is presented in Table 2. In this analysis, we excluded race missing data cases. Although we can see in the Table 1, this category is similar to non indigenous.

In the multivariate analysis, the following variables were associated to indigenous: male sex (OR = 0.83; CI 95%: 0.82–0.84); patients younger than 1 year in indigenous populations (OR = 1.91; CI 95%: 1.80–2.03), whose OR decreased as age increased, at least up to 40 years; education has greater magnitude among illiterates (OR = 1.31; CI 95%: 1.28–1.35), with reduction of OR as the education level increased; active case surveillance was strongly associated to indigenous malaria (OR = 1.73; CI 95%:1.70–1.75); the highest parasitemia (++++, i.e. >10 parasites 2.73 per one thick film field) is associated with malaria in the indigenous population (OR = 2.10; CI 95%:1.53–2.88); *P. vivax*, mixed infections (up to twice as many cases in indigenous patients) (OR = 2.42; CI 95%:2.2–2.66) and other, i.e., infections caused by *P. malariae* and *P. ovale* were 4 times more common in cases of malaria in the indigenous group (OR = 4.82; CI 95%: 2.49–9.35); timeliness of treatment, longer than 48 hours, appears as a malaria protective factor in indigenous (OR = 0.98; CI 95%: 0.97–1.00) (Table 2).

Regarding malaria mortality in the period from 2007 to 2016, 109 deaths were reported (Table 3). Of these, two notifications were excluded due to lack of information or errors when filling in the forms. Considering indigenous groups, there were 34 deaths, 13 female and 21 male. Higher mortality levels were observed in those older than 60 years (32.3%). Regarding the non-indigenous group, there were 73 deaths, the highest number of deaths among female patients, and the most prevalent age group was between 20 and <40 years of age.

Both categories (Indigenous and Non-indigenous) had underlying cause of death—ICD (International Statistical Classification of Health-Related Diseases and Problems) of death reported showing a greater proportion of *P. vivax* malaria.

**Table 2. Factors associated with malaria (crude and adjusted analysis), according to indigenous patients (village or self-reported indigenous) or non-indigenous.** Amazonas state, Brazil, 2007–2016.

| Variables | Indigenous* | | | |
|---|---|---|---|---|
| | Crude OR | CI 95% | Adjusted OR | IC 95% |
| **Sex** | | | | |
| Female | 1.0 | | 1.0 | |
| Male | 0.77 | 0.76–0.78 | 0.83 | 0.82–0.84 |
| **Age group (years)** | | | | |
| < 1 | 2.69 | 2.54–2.84 | 1.91 | 1.80–2.03 |
| 1 to < 10 | 2.03 | 1.97–2.11 | 1.62 | 1.56–1.68 |
| 10 to < 20 | 1.11 | 1.08–1.15 | 1.58 | 1.52–1.64 |
| 20 to < 40 | 0.89 | 0.62–0.92 | 1.34 | 1.29–1.39 |
| 40 to < 60 | 0.71 | 0.68–0.73 | 0.93 | 0.89–0.96 |
| 60 or over | 1.0 | | 1.0 | |
| **Schooling (years)** | | | | |
| Illiterate | 1.04 | 1.02–1.07 | 1.31 | 1.28–1.35 |
| 1 to 4 | 0.38 | 0.37–0.39 | 0.47 | 0.46–0.48 |
| 5 to 9 | 0.28 | 0.27–0.29 | 0.38 | 0.36–0.39 |
| 10 or more | 0.29 | 0.28–0.30 | 0.4 | 0.41–0.44 |
| Not applicable | 1.0 | | 1.0 | |
| **Type of surveillance slide** | | | | |
| Passive case surveillance | 1.0 | | 1.0 | |
| Active case surveillance | 1.96 | 1.93–1.98 | 1.73 | 1.70–1.75 |
| **Parasitemia level by plus system scale** | | | | |
| (completeness: 99.5%) | | | | |
| <+/2 (less than one parasite per 100 thick film fields) | 1.0 | | 1.0 | |
| +/2 (up to one parasite per 100 thick film fields) | 1.49 | 1.47–1.52 | 1.62 | 1.59–1.65 |
| + (1 to 10 parasites per 100 thick film fields) | 1.45 | 1.42–1.47 | 1.64 | 1.61–1.67 |
| ++ (11 to 100 parasites per 100 thick film fields) | 1.34 | 1.32–1.36 | 1.53 | 1.50–1.55 |
| +++ (1 to 10 parasites per one thick film field) | 1.96 | 1.87–2.06 | 1.80 | 1.71–1.90 |
| ++++ (>10 parasites per one thick film field) | 2.73 | 2.03–3.68 | 2.10 | 1.53–2.88 |
| **Results of the test** | | | | |
| *P. vivax* | 1.76 | 1.73–1.80 | 1.61 | 1.58–1.65 |
| *P. falciparum* | 1.0 | | 1.0 | |
| Mixed infection | 3.05 | 2.80–3.32 | 2.42 | 2.21–2.66 |
| (*P. falciparum* + *P.vivax*) | | | | |
| Other** | 3.27 | 1.73–6.20 | 4.82 | 2.49–9.35 |
| **Timeliness of treatment (hours)** | | | | |
| ≤ 48h | 1.0 | | 1.0 | |
| > 48h | 1.03 | 1.02–1.04 | 0.98 | 0.97–1.00 |

Source: Epidemiological Surveillance Information System for Malaria, data obtained January, 2018.

OR: odds ratio; 95% CI: 95% confidence interval.

Significance level <0.20 for simple logistic regression; Significance level <0.05 for multivariate logistic regression.

P-value was significant in all variables and categories.

*Indigenous classification is the combination of an individual with a declared indigenous status or having the indigenous village registered as the place where infection occurred.

** Other = *P. ovale*, *P. falciparum* + *P. malariae* or *P. malariae*.

**Table 3. Distribution of deaths caused by malaria, in indigenous and non-indigenous populations, Amazonas state, 2007 to 2016.**

| Variables | Indigenous | | Non-indigenous | |
|---|---|---|---|---|
| | n | % | n | % |
| **Sex** | | | | |
| Female | 13 | 38.24 | 40 | 54.79 |
| Male | 21 | 61.76 | 33 | 45.21 |
| **Age group (years)** | | | | |
| < 1 | 4 | 11.76 | 3 | 4.11 |
| 1 to < 10 | 6 | 17.65 | 13 | 17.81 |
| 10 to < 20 | 5 | 14.71 | 5 | 6.85 |
| 20 to < 40 | 5 | 14.71 | 21 | 28.77 |
| 40 to < 60 | 3 | 8.82 | 12 | 16.44 |
| 60 years or over | 11 | 32.35 | 19 | 26.03 |
| **Underlying cause of death (ICD)** | | | | |
| B509 –Malaria caused by *Plasmodium falciparum* with unspecified severity | 6 | 17.65 | 14 | 19,18 |
| B519 –Malaria caused by *Plasmodium vivax* without complications | 12 | 35.29 | 29 | 39.73 |
| B54—Malaria caused by unidentified parasite | 9 | 26.47 | 16 | 21.92 |
| B518—Malaria caused by *Plasmodium vivax* with other complications | 4 | 11.76 | 11 | 15.07 |
| B500—Malaria caused by plasmodium *falciparum* with cerebral complications | 1 | 2.94 | 3 | 4.11 |
| B538 –Other types of malaria with parasitological confirmation, unclassified in other sections | 1 | 2.94 | - | 0.00 |
| B508 –Other severe and complicated types of malaria caused by *Plasmodium falciparum* | 1 | 2.94 | - | 0.00 |

Source: Mortality Information System, data obtained in January, 2018.

ICD (International Statistical Classification of Health-Related Diseases and Problems).

## Discussion

In the state of Amazonas, malaria is a disease that predominates in rural populations and reflects social and economic aspects that are accentuated in indigenous populations. Here the authors shed light on the factors associated with the occurrence of malaria in indigenous populations and cases that occurred in indigenous villages, as well as the description of the profile of deaths recorded by malaria in these populations in the state of Amazonas, by analyzing a period of 10 years (2007 to 2016).

The predominance of malaria in indigenous males is similar to the results found in other studies conducted in the Amazon Region [2, 11–14], which justify such predominance based on indigenous behavior and a lifestyle based on extractivism and family farming, which is predominantly performed by men who consequently are more exposed to contact with the vector and the parasite [15, 16].

The highest proportion of cases among children up to 10 years old in indigenous peoples found in this study is equivalent to historical studies in the Amazon [17, 18]. In a riverine community in Portochuelo (Rondônia), a higher prevalence of malaria was found in children under 16 years of age [19]. Similar results were found in a study with the Yanomami tribe, on the border between Brazil and Venezuela, where children under 16 were the most affected by malaria [20]. In other studies, results were also found in younger age groups (children under 14 years old), such as schoolchildren and preschoolers [11, 21, 22].

The malaria treatment guideline envisages immediate malaria treatment (timeliness of treatment) to be provided to all patients with positive slide results or positive rapid test results for both symptomatic and asymptomatic cases [23]. This fact may be closely related to the high proportion of cases with time of up to 48 hours between onset of symptoms and time of

diagnosis and/or treatment time of up to 48 hours in indigenous people. A study conducted in the state of Amazonas, with a non-indigenous population, showed that most patients were diagnosed predominantly within three days of symptom onset (94.3%). The timeliness of treatment diagnosis and treatment helps to prevent hospitalizations and deaths, but also help to control disease transmission by preventing or reducing the onset of parasite (gametocyte) sexual stages in human hosts, which are the infectious forms in the vectors [24].

Regarding the *Plasmodium* species, the present study showed similar results to those performed in non-indigenous populations, if we consider that *P. vivax* account for majority of the cases [25, 26]. An epidemiological malaria survey found prevalence of *P. vivax* cases, followed by *P. falciparum* and mixed infection cases (*P. vivax* + *P. falciparum*) [27]. It is necessary to remark that we found highest prevalence of P. falciparum and mixed infection in indigenous population, which cause more impact in the indigenous in terms of severity fo the disease. Maybe this aspect can explain that in indigenous we had more cases in the passive detection, which leads people to look for diagnosis faster because of the symptoms.

In Brazil, the incidence of *P. falciparum* and *P. vivax* cases occurred at similar levels until 1988, after which the proportion of *P. falciparum* cases progressively decreased, while cases involving *P. vivax* increased and from then onwards became the predominant species responsible for over 90% of episodes of malaria [4, 5, 27]. It is important to note that there are biological differences between *P. vivax* and *P. falciparum*, which make *P. vivax* particularly challenging to control and eliminate in malaria endemic regions [28–30]. According to a study in Papua New Guinea, relapses may be the cause of about 80% of *P. vivax* cases [31]. A recent study in the state of Amazonas also suggests that relapses made a significant contribution to cases in this region [32].

The Timeliness of treatment found in this study can be interpreted as an effort to advance treatment initiation is due to the expansion of the laboratory network and the action of community health workers (CHA), which contributed to the decreasing trend observed in the percentage of malaria hospitalization of 3.3% in 1999 to 1.4% in 2009: a significant reduction of 79% [3].

The biggest obstacles to malaria treatment in indigenous populations is getting access to them due to the population dispersion and also the great distances involved as well as the local geography which both make the logistics of care more expensive. From the perspective of the diagnosis and treatment of malaria in indigenous peoples, the greatest difficulty is in reaching remote villages, due to the complexity of access which is only by river. The microscopy network does still not meet the total population coverage. The performance of well-trained and active indigenous health agents (IHAs) is a positive aspect for disease control [13].

In a study on factors associated with malaria treatment in the Brazilian Amazon, it was found that timeliness of treatment was associated with indigenous patients with low education (from zero to 5 years of schooling). These variables indicate vulnerable and highly dependent groups of the public Unified Health System (SUS) [23].

In endemic areas in Brazil, people are aware of the symptoms of malaria, and usually seek care early on in the clinical manifestations, which makes severe conditions uncommon. A survey conducted in the state of Acre, Brazil found that about 22% of patients sought care in less than 48 hours, while more than 70% of people looked for care less than four days from the onset of symptoms and most patients had malaria by *P. vivax* [33]. These are probably the explanations for finding only non-severe cases in this casuistry. This early search for diagnosis and treatment is also important to prevent people with protozoa in the peripheral circulation from acting as a source of infection for the disease transmitter, as proper treatment eliminates the blood parasite within a few days. This may shed light on the importance of timeliness of treatment and low mortality from malaria.

The adjusted multivariate analysis showed male is a protective factor for indigenous people, which differs from general population. This finding deserve investigation, since malaria transmission is related to activities that promote contact with vector, we do not know what occurs with indigenous.

Regarding age, the strength of the association of malaria in indigenous people is greater the smaller the age group. This fact corroborates another study in indigenous populations, in which the disease is associated with children [20]. On the other hand, it differs from other studies that detect a greater problem of malaria in the economically active population among non-indigenous people [23]. In the case of indigenous people, transmission may be intra- or peridomestic, where the vector is more likely to infect children, especially when they sleep.

The educational level must be interpreted with caution, since the indigenous populations, as they live in areas far from urban centers, may have more difficulty in accessing school. Perhaps this explains the greater chance of being a case among the illiterate.

The chance of malaria cases among indigenous people through the active case surveillance may result from the way in which indigenous health services are organized, in systematically promoting this active case surveillance. If we combine the interpretation with the timeliness of treatment, it is observed that even though the majority occurring within 48 hours, when it is more than 48h, it was a protective factor. Some explanation about the organization of malaria control actions in the Indigenous population can be found in a study on the municipality of São Gabriel da Cachoeira, which has the most indigenous people concentration in state of Amazonas [34].

The level of parasitemia can be an indicator of the severity of the disease among indigenous people, with greater strength of association in cases of highest parasitemia (++++). Still on gravity, the parasitic form has a stronger association for mixed infection (*P. falciparum + P. vivax*) and another parasitic form (*P. ovale*, *P. falciparum + P. malariae* or *P. malariae*), which reveals that indigenous people have particularities to be considered, since whereas in the state of Amazonas and in non-indigenous populations the predominance is of *P. vivax*.

The malaria mortality must be a rare event since the diagnosis and the treatment is freely available, and the health professionals are conscious of the importance in the case finding and management. However the mortality analysis reveals the unfavorable outcome, specially in male and older people. The underlying cause of death was caused by *P. vivax* with no complications in both population groups, indigenous and non indigenous. One problem with *P. vivax* infection is the possibility of causing unfavorable and unmeasured outcome [9, 35].

The limitations of this study are mainly related to the use of secondary data itself. For example, the lack of information on job occupation or other variable proxy to socioeconomic status did not allow to include in the analysis in order to find a better model adjustment. However, it is not possible to control report, missing data or typing errors, and also the underreporting malaria cases. The last cited can be a minor problem, since it is a compulsory notification disease, and the treatment is just made available after case notification. To tackle with race missing data, we proposed this method of race data imputation, which revealed to be useful to conduct this analysis. However it is known that, ideally, the 100% of completeness must be the goal of malaria control program.

Even considering the problems of malaria secondary data, we emphasize enhancing of the quality, its universal coverage in Brazil, the availability of great data amount and the possibility to work with indigenous data, with no need to go to field data collection, which in indigenous issues has other than costs implications.

Despite these limitations, we judge that not compromise the findings. They are useful in understanding of the malaria peculiarities in the indigenous population, since we found differences in the associated factors. We also remark the importance of the completeness of

secondary data, mainly the variable race, because it can reveals cases profile, cases inequalities and contribute for malaria control.

## Conclusions

The study findings presented here have identified the factors associated with malaria in indigenous populations and in other races. Among indigenous people, the disease predominates in men of a younger age and who have high parasitemia. Because indigenous inhabit places that with difficult access and with few health resources, the disease is more often detected via active case surveillance. The border municipalities which have indigenous populations presented higher than average percentage of malaria cases. Evidently there is a clear need to work on actions related to the planning and execution of health care for these locations.

Obviously neglected, indigenous people in the Amazon region are particularly affected by severe infectious diseases such as malaria, which adds an obstacle to both the quality of life of these populations and the advance of disease control and elimination in the region. The findings presented here clarify the malaria transmission profile in these populations, as well as their risk factors, generating evidence that supports decision-making by managers of the Malaria Control Programs in the Amazon Region.

## Supporting information

**S1 Table. Distribution of cases of malaria by municipality.** Amazonas state, Brazil, 2007 to 2016.
(DOCX)

## Author Contributions

**Conceptualization:** Wuelton Marcelo Monteiro, Maria Jacirema Ferreira Gonçalves.

**Data curation:** Bruna Martins Meireles, Wuelton Marcelo Monteiro, Maria Jacirema Ferreira Gonçalves.

**Formal analysis:** Bruna Martins Meireles, Wuelton Marcelo Monteiro, Maria Jacirema Ferreira Gonçalves.

**Methodology:** Bruna Martins Meireles, Vanderson de Souza Sampaio, Wuelton Marcelo Monteiro, Maria Jacirema Ferreira Gonçalves.

**Supervision:** Maria Jacirema Ferreira Gonçalves.

**Writing – original draft:** Bruna Martins Meireles, Vanderson de Souza Sampaio, Wuelton Marcelo Monteiro, Maria Jacirema Ferreira Gonçalves.

**Writing – review & editing:** Bruna Martins Meireles, Vanderson de Souza Sampaio, Wuelton Marcelo Monteiro, Maria Jacirema Ferreira Gonçalves.

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
