## [Decision Letter · Decision Letter 0]

18 Jun 2020

PONE-D-20-05382

Factors associated with malaria in indigenous populations: a retrospective study from 2007 to 2016

PLOS ONE

Dear Dr. Gonçalves,

Thank you for submitting your manuscript to PLOS ONE.  Your submission has now been peer-reviewed by three experts in the field and myself. I agree that the manuscript would benefit from being revised according to the suggestions following and encourage you to do so. Therefore, we invite you to submit a revised version of the manuscript that addresses the points raised during the review process.

ACADEMIC EDITOR: Herein the authors aim to identify the factors related to malaria in indigenous people in Amazon sate, Brazil, from 2007 to 2016. To accomplish this, they did use the Brazillian Malaria Epidemiological Surveillance Information System and the Mortality Informative System. They found that sex, age, education level, malaria surveillance system, parasitemia level, plasmodium species, and time to treatment were strongly associated with malaria in the indigenous group.

Overall the manuscript is well written and organized. But several pitfalls do exist regarding the methodology.

General:

Please rewrite the abstract as suggested by one of the reviewers

Perform  language copyediting for the entire text

Methods

Provide sufficient details about the SIVEP, SIVEP-Malaria, and MIS. Similarly, contextualize the readers your study setting, specifically the nuances about the Amazon state, the tribal villages, the national malaria program guidelines. How community healthcare workers perform the diagnosis of malaria in indigenous people? Any surveillance underway?

How did you handle missing data?

Results

Does the outcome and severity of malaria infection vary according to race? Could you present malaria incidence by year? Is there any variation of malaria incidence in indigenous people throughout the study period?

Discussion

Address the limitations of your findings considering the drawbacks of working with national epidemiological information systems

We look forward to receiving your revised manuscript.

Kind regards,

José Moreira, MD, MSc

Academic Editor

PLOS ONE

Journal Requirements:

2. Please ensure you have thoroughly discussed all potential limitations of this study within the Discussion section.

3. We note that [Figure(s) 1] in your submission contain [map/satellite] images which may be copyrighted. All PLOS content is published under the Creative Commons Attribution License (CC BY 4.0), which means that the manuscript, images, and Supporting Information files will be freely available online, and any third party is permitted to access, download, copy, distribute, and use these materials in any way, even commercially, with proper attribution. For these reasons, we cannot publish previously copyrighted maps or satellite images created using proprietary data, such as Google software (Google Maps, Street View, and Earth). For more information, see our copyright guidelines: http://journals.plos.org/plosone/s/licenses-and-copyright.

1.    You may seek permission from the original copyright holder of Figure(s) [1] to publish the content specifically under the CC BY 4.0 license. 

Reviewers' comments:

Reviewer's Responses to Questions

**Comments to the Author**

1. Is the manuscript technically sound, and do the data support the conclusions?

Reviewer #1: Partly

Reviewer #2: Partly

Reviewer #3: Yes

2. Has the statistical analysis been performed appropriately and rigorously? 

Reviewer #1: Yes

Reviewer #2: Yes

Reviewer #3: Yes

3. Have the authors made all data underlying the findings in their manuscript fully available?

Reviewer #1: Yes

Reviewer #2: No

Reviewer #3: Yes

4. Is the manuscript presented in an intelligible fashion and written in standard English?

Reviewer #1: Yes

Reviewer #2: Yes

Reviewer #3: No

5. Review Comments to the Author

Reviewer #1: The authors chose an interesting subject, with few approaches on literature regarding indigenous malaria in Brazil, but needs some revision concerned the following aspects:

1) Developing the indigenous lifestyle that contributed to presented results

2) In Methodology the paragraph about including as indigenous people those which referred living in "aldeias" it was not clear the meaning of the sentence " The only possible cases of erroneous inclusion would

be the small number of health professionals who access such places to provide indigenous

health care. The authors should explain what they had on mind with this assumption.

3) In Methodology - Definition of active and passive case malaria surveillance should be provided in methodology

4) In Methodology - It should be well defined how many thick blood smears were considered to establish malaria diagnosis since "the number of parasites in thick smears was graded using the plus system scale:"

5) In Methodology - please provide the number and the data of the approval of the study by the Ethics Review Board of the Federal University of Amazonas.

6) The authors should provide percentages of some observed variables such as age, parasitemia to better charactherize their found results.

7) Necessity of reviewing table 1 concerning some variables that do not match the mentioned percentage on parenthesis; the absolute values of some variables do not reach 100%, being low or above this percentage.

8) Please, add in Fig 1 the percentages observed, instead of informing only colors in the polygons -dark and light blue, respectively to low and high percentages

9) To strength the discussion (and even the conclusion) about the biggest obstacles to malaria treatment in indigenous populations that is getting access to them due to their geographical location as well as their dispersion in the state of Amazon, authors should include a map with this information on the results

10) Based on the results, two topics in the conclusion need to be better addressed in the discussion - the reason why indigenous peoples of the state of the Amazon is "obviously" neglected and the significant predominance of passive case malaria diagnosis.

Reviewer #2: The authors present malaria burden in the Amazonas in the last decade and identify the factors associated with it. However, the authors need to address certain points before the manuscript is suitable for publication.

Page 4

1. Data of cross-sectional studies needs further explanations (how data was collected, target population, ACD vs PCD...). This is a key aspect that requires visibility in the manuscript.

2. A paragraph explaining a socio-economic description of the state of Amazonas (race distribution, wealth), and the epidemiology of malaria in the state is missing (access to malaria prevention, first-line treatment, vivax vs falciparum, ...).

Page 5:

3. Which diagnostic test was used for confirmation of malaria? Was it performed at health facility or actively in the community? If only microscopy was used it must be stated as limitation in the paragraph were low RDT usage is discussed.

4. Expand the multiple imputation methods in an appendix.

Page 6:

5. Variables included: what about malaria prevention (ITNs and IRS), and women being pregnant? Do the authors think that missing to include these variables might have an impact on the results?

6. I understand that job occupation and wealth (or socio-economic status) were not collected. If not available, they need to be included in study limitations in the discussion.

Page 7:

7. It is briefly discussed at the end of the manuscript, but did the authors address the issue of malaria relapses associated with P. vivax? If done, how were relapses excluded/differentiated?

Page 8:

8. It is important that the authors clarify the terms active case detection and passive case detection in the methods, and include further interpretations in the discussion.

Page 12:

9. The first two paragraphs can be combined to avoid repetition.

Discussion section:

10. This section needs a limitation paragraph and address potential biases of the study.

Reviewer #3: The aim of this study was to identify factors associated with malaria in indigenous people, in Amazonas state, Brazil. The study used secondary data obtained from the Brazilian malaria surveillance program.

Some major comments and minor details that require further information or consideration:

Abstract

- There is no sentence about background, the first phrase is the objective without basic information of where, when, why.

- The period is repeated in Methods and Results.

- The study design is not described.

- There is no information about both data systems are Brazilian systems.

- The basic statistical analysis is not too important to describe in the abstract. Definitions or study local details or variables is more important than those.

- The phrase: “This state has the highest proportion of indigenous peoples and malaria in Brazil” in the Results section is a result of the study or an information about the study site? If it is an information should be in the Method section or in Background.

- There is no results about the factors cited as associated malaria and indigenous people (measures of association, 95% IC, p).

- The conclusion is not in accordance with the results presented in the abstract.

Introduction

- There is no dot in the end of the second sentence.

- In the sentence: “The production and transmission of malaria”, the “production of malaria” could be replaced by development or occurrence.

- Page 4: last sentence: The sentence would better written as: The aim of this study was to identify factors associated with malaria in indigenous people.

Methods

- The order of the sub-sections could be changed to better understand the methodology. First of all, it is necessary to describe the study site, study population, study design and ethical considerations. After that, describe the procedures, the data analysis, the definitions of variables.

- The study design did not mention the use of secondary data.

- “Definitions” sub-section is related to “Study population”.

- The Study location could have more detailed. For instance, how many tribes/villages (aldeias), how many malaria screenings sites, amount of the indigenous population.

- The sentence “The database was organized and analyzed using Stata software version 13” could be relocated to Data analysis sub-section.

- To make sense I suggest to better describe the procedures, step by step. First, from where the data were obtained, describe the surveillance systems in Brazil, the notification case form. Then, the selection of variables and definitions.

Results

-Page 8: 1st paragraph: “Therefore, cases without race information were reclassified for the purpose of this study. These were classified as indigenous when they had their indigenous race registered or because they were a case where the probable place of infection was the indigenous village. Thus, 42,702 subjects, who previously had missing information on race, received an indigenous classification based on this criterion.” If they had their indigenous race registered so they were not a case without race. Those sentences should be rewritten. First describe how indigenous was identified. Then, describe what you did in the situation of cases without race information. Those sentences could be relocated to Methods section.

- Page 8: There are no results in second, third and fourth paragraphs. They are summary of results. Summary of results is better located in paragraphs of Discussion section to introduce the discussion of the results. In the Results section it is necessary write the results, reporting n as well as %, measures of association, 95% IC, p. If a descriptor is used in the sentence (e.g. "the largest percentage") include number and percentage in brackets (n=#, %).

- Page 12: the first paragraph is different from the second paragraph. In the first one there is no result described, in the second one there are for some variables.

- Page 12: second paragraph: is better to describe as variables associated with indigenous population, not as significant variables.

- Page 12: second paragraph: some variables has results in brackets, some has not. Describe all.

- Page 12: second paragraph: “parasitemia in plus system scale is higher in the last strata of the scale” does not make sense. What does represent? Describe as higher parasitemia is associated with malaria in indigenous population.

- Table 2, page 14: describe “Other” Results of the Test, in the table legend, for example.

Discussion

- Page 16, last paragraph: about what results the paragraph is related? The paragraph could be relocated to Study location sub-section.

- Page 17: “After diagnosis of malaria, treatment was immediately prescribed and administered free of charge to all the patients [25].” This statement is redundant. It could be relocated to Study location sub-section, to better illustrate the Brazilian malaria program. It is important to briefly comment about indigenous malaria health program, if is apart of Brazilian general health program or not.

- Page 17: “Regarding the Plasmodium species, the present study showed similar results to those performed in non-indigenous populations, with predominance of P. vivax as an infectious agent [27, 28]. An epidemiological malaria survey found prevalence of P. vivax cases, followed by P. falciparum and mixed infection cases (P. vivax + P. falciparum) [29], a result similar to that found in our study.” About these affirmations, the results showed the prevalence of P. falciparum in indigenous population was 13.09% while the prevalence in non-indigenous was 7.92%. For mixed infection, the prevalence in indigenous population was 0.93% and non-indigenous was 0.33%. We know P. falciparum and mixed infection P.falciparum/P.vivax are different in terms of morbidity, mortality, costs etc. P. vivax is the predominant species in both population but there is a difference in the distribution of species and consequently likely in the burden of disease in the indigenous population.

- There are no considerations about limitations of the study.

- The factors identified as associated with malaria in indigenous populations (table 2) were not discussed in the Discussion section.

- The results of table 3 were not discussed in the Discussion section.

- Sometimes they describe “the state of Amazonas” and “Amazon” for the apparently the same place.

General comments

- Add line numbers in the document helps us to point the text.

- I suggest the authors to follow the STROBE guidelines on reporting of observational studies.

- It would benefit from additional editing for English language and for redaction.

6. PLOS authors have the option to publish the peer review history of their article (what does this mean?). If published, this will include your full peer review and any attached files.

Reviewer #1: No

Reviewer #2: No

Reviewer #3: No

---

## [Author Response · Author response to Decision Letter 0]

21 Aug 2020

As requested, we provide in this submission an entirely reviewed version of our manuscript, and send attached the editor and reviewers' response

---

## [Decision Letter · Decision Letter 1]

2 Oct 2020

Factors associated with malaria in indigenous populations: a retrospective study from 2007 to 2016

PONE-D-20-05382R1

Dear Dr. Gonçalves,

Your manuscript has now been formally accepted for publication in PLoS One. Please see important details concerning the publication process below.

Your efforts during the process of revision are acknowledged and I hope you also are pleased with the final result.

We appreciate being able to publish your work and look forward to seeing your paper online as soon as possible.

Kind regards,

José Moreira, MD, MSc

Academic Editor

PLOS ONE

Additional Editor Comments (optional):

Reviewers' comments:

Reviewer's Responses to Questions

**Comments to the Author**

1. If the authors have adequately addressed your comments raised in a previous round of review and you feel that this manuscript is now acceptable for publication, you may indicate that here to bypass the “Comments to the Author” section, enter your conflict of interest statement in the “Confidential to Editor” section, and submit your "Accept" recommendation.

Reviewer #1: All comments have been addressed

Reviewer #2: All comments have been addressed

Reviewer #3: All comments have been addressed

2. Is the manuscript technically sound, and do the data support the conclusions?

Reviewer #1: Yes

Reviewer #2: Yes

Reviewer #3: Yes

3. Has the statistical analysis been performed appropriately and rigorously? 

Reviewer #1: Yes

Reviewer #2: Yes

Reviewer #3: Yes

4. Have the authors made all data underlying the findings in their manuscript fully available?

Reviewer #1: Yes

Reviewer #2: Yes

Reviewer #3: No

5. Is the manuscript presented in an intelligible fashion and written in standard English?

Reviewer #1: Yes

Reviewer #2: No

Reviewer #3: Yes

6. Review Comments to the Author

Reviewer #1: The manuscript fulfilled the recommendations we had suggested and it is accepted for publication in PLos One

Reviewer #2: The authors have addressed the comments suggested. However, the article will highly benefit from an in-detail revision of the English grammar.

Reviewer #3: The authors have adequately addressed my comments. Some one minor detail that require correction:

Introduction

- Please correct the last sentence: “The aim of this study was to The aim of this study was to...”

7. PLOS authors have the option to publish the peer review history of their article (what does this mean?). If published, this will include your full peer review and any attached files.

Reviewer #1: No

Reviewer #2: No

Reviewer #3: No

---

## [Editor Report · Acceptance letter]

12 Oct 2020

PONE-D-20-05382R1 

Factors associated with malaria in indigenous populations: a retrospective study from 2007 to 2016 

Dear Dr. Gonçalves:

I'm pleased to inform you that your manuscript has been deemed suitable for publication in PLOS ONE. Congratulations! Your manuscript is now with our production department. 

Kind regards, 

on behalf of

Dr. José Moreira 

Academic Editor

PLOS ONE